# Enzymatic Assessment of the State of Oil-Contaminated Soils in the South of Russia after Bioremediation

**DOI:** 10.3390/toxics11040355

**Published:** 2023-04-08

**Authors:** Tatyana Minnikova, Sergey Kolesnikov, Sofia Revina, Anna Ruseva, Vladimir Gaivoronsky

**Affiliations:** 1Department of Ecology and Nature Management, Academy of Biology and Biotechnology by D.I. Ivanovsky, Southern Federal University, Stachki Ave., 194/1, Rostov Region, 344090 Rostov-on-Don, Russia; kolesnikov1970@list.ru (S.K.); soffy.soff@mail.ru (S.R.); ruseva.ann@yandex.ru (A.R.); 2Academy of Physical Culture and Sports, Department of Theoretical Foundations of Physical Education, Southern Federal University, St. Zorge, 5, Rostov Region, 344015 Rostov-on-Don, Russia; vggayvoronskiy@sfedu.ru

**Keywords:** soil, pollution, biodiagnostics, activity of catalase, activity of dehydrogenases, activity of invertase, activity of urease, activity of phosphatase

## Abstract

Soil pollution with oil as a result of accidents at oil pipelines and oil refineries is a frequent occurrence in the south of Russia. To restore such polluted lands, it is necessary to carry out soil remediation measures. This work aimed to evaluate the use of ameliorants of various natures (biochar, sodium humate, and microbial preparation Baikal EM-1) to restore the ecological state of oil-contaminated soils with different properties (Haplic Chernozem, Haplic Arenosols, Haplic Cambisols). To assess the ecological state of soils, the following physicochemical and biological indicators were studied: residual oil content, redox potential, and medium reaction (pH). Changes in enzymatic activity were also studied, including catalase, dehydrogenases, invertase, urease, and phosphatase. The greatest decomposition of oil in Haplic Chernozem and Haplic Cambisols was provided by Baikal EM-1 (56 and 26%), and in Haplic Arenosols, this was provided by biochar (94%) and sodium humate (93%). In oil-contaminated Haplic Cambisols, the content of easily soluble salts with the addition of biochar and Baikal EM-1 increased by 83 and 58%, respectively. The introduction of biochar caused an increase in pH from 5.3 (Haplic Cambisols) to 8.2 (Haplic Arenosols). The introduction of oil-contaminated Haplic Arenosols of biochar, humate, and Baikal stimulated the activity of catalase and dehydrogenases by 52–245%. The activity of invertase was stimulated in the Haplic Chernozem after the introduction of ameliorants by 15–50%. The activity of urease was stimulated after the introduction of ameliorants into borax and Arenosol by 15–250%. The most effective ameliorant for restoring the ecological state of Haplic Cambisols after oil pollution was biochar. For Haplic Arenosols, this was sodium humate, and for Haplic Chernozem, the effectiveness of biochar and sodium humate did not differ. The most informative indicator for the remediation of Haplic Chernozem and Haplic Cambisols was the activity of dehydrogenases, and for Haplic Arenosols, this was the activity of phosphatase. The results of the study should be used to biomonitor the ecological state of oil-contaminated soils after bioremediation.

## 1. Introduction

Every year, millions of hectares of soil in the world, including agricultural land, are contaminated with oil and petroleum products, which leads to the deterioration of the biological condition of soils, a decrease in their fertility, and, therefore, the violation of agricultural and ecological functions [1,2,3,4,5,6,7]. Currently, various methods of bioremediation have been developed to eliminate soil contamination with oil and petroleum products. Substances of different natures and mechanisms of action are used as bioremediants, including microbiological preparations, organic and mineral fertilizers, mineral sorbents, and others [4,5,8]. When assessing the effectiveness of ameliorants, the main category of assessment is the residual oil content, and the restoration of the biological properties of the soil remains poorly studied. To assess the ecological state of soil during oil pollution, as with other anthropogenic impacts, the most informative are biological indicators that quickly respond to external influences and correlate with the concentration of petroleum hydrocarbons in the soil. After the introduction of ameliorants, it is necessary to use the most informative and sensitive indicators of biodiagnostics of oil-contaminated soils, making it possible to assess the effectiveness of their restoration by the rate of restoration of soil fertility. Currently, many restoration technologies are available to work with soils contaminated with petroleum hydrocarbons, including solvent washing followed by extraction, incineration and thermal desorption, chemical oxidation, electrokinetic remediation, and other approaches [9,10,11,12,13]. However, in most of these methods, there is no possibility of mass application, due to the high cost and the threat of secondary pollution with petroleum hydrocarbons. Therefore, it is a priority to use alternative technologies of soil restoration that enhance the process of pollutant degradation [13,14,15,16,17,18].

The decomposition of petroleum hydrocarbons is influenced by oxygen availability (soil air regime), the reaction of the soil environment, temperature, redox potential, moisture content, organic carbon, and other biogenic elements, and the composition of microbial communities. It is known that high temperatures and high content of easily soluble salts enhance the effects of the inhibition of microbial communities and the inhibition of soil enzymes when contaminated with petroleum hydrocarbons [19]. In addition, the decomposition rate depends on the composition of petroleum hydrocarbons and the presence of associated pollutants [13,20,21].

The aim and objective of this study was to assess the biological state of oil-contaminated soils in the South of Russia after bioremediation. To achieve this aim, the following tasks are solved: (1) to evaluate the effect of ameliorants on the oil content in each type of soil; (2) to analyze the change in physico-chemical parameters in each type of soil; (3) to enhance the change in enzymatic activity in each type of soil; and (4) to compare the remediation efficiency of each ameliorant for each soil type.

## 2. Materials and Methods

The objects of the study were three soils differing in their physical and chemical properties (Table 1). The most fertile soils in the world include the Haplic Chernozem of Southern Russia. Their restoration and preservation are especially important given the huge anthropogenic load on these soils.

Samples of Haplic Chernozem were selected in the territory of the Rostov region. Ordinary clay loam chernozem, i.e., Haplic Chernozem according to WRB (2022), was selected as the first object of the study [22]. The place of selection was the arable land of the Botanical Garden of the Southern Federal University (A horizon 0–10 cm) in the center of Rostov-on-Don (47°14′17.54″ N; 39°38′33.22″ E).

The second object of the study was light loam ordinary chernozem, i.e., Haplic Arenosols [22]. Soil samples were taken from arable land (A horizon 0–10 cm) in the Ust’-Donets district of the Rostov Region (47°46.015 N; 40°51.700 E).

The third object of research was brown forest clay loam soil, i.e., Haplic Cambisols [22]. Soil samples were taken in a hornbeam-beech forest (A horizon 0–10 cm) near the Nikel village, the Republic of Adygea (44°10.649 N; 40°9.469 E).

In this study, the stability of Haplic Chernozem was compared with Haplic Cambisol and Haplic Arenosols. 

Oil provided by the Novoshakhtinsk’s oil Refinery (Novoshakhtinsk) was used to simulate pollution. Oil with a density of 0.861 kg/m^3^, sulfur content of 1.34%, water mass of 0.27%, concentration of chloride salts of 73 mg/dm^3^, mass fraction of mechanical impurities of 0.006%, and mass fraction of paraffin of 4.46% were used for soil contamination in this model experiment. Oil was introduced into each vessel with soil at the rate of 5% of soil mass.

Biochar, sodium humate, and Baikal EM-1 were used to restore soil functions after oil pollution. These drugs affect oil decomposition by biostimulation and adsorption.

Biochar (grade A) is pure charcoal with a carbon content of at least 85%, a density equal to 0.37 g/cm^3^, ash mass fraction of 3%, and water mass fraction of no more than 6% (GOST 7657-84). In this study, biochar produced by LLC DianAgro was used (Novosibirsk, Russia). The product was produced by pyrolysis of birch wood (*Betula alba*) at a temperature of 360–380 °C, without oxygen access, on retort installations. For remediation, 5% of the biochar from the soil mass was introduced into the soil.

Sodium humate is an organo-mineral fertilizer containing 70% humic acid salts, which are one of the most important sources of nutrition for native soil microbiota. From trace elements, sodium humate contains molybdenum, copper, cobalt, manganese, and zinc, and from heavy metals, lead and cadmium were obtained. The dry composition contains nitrogen, phosphorus, potassium, magnesium, and calcium. Sodium humate, due to the presence of sodium ions in Haplic Chernozem, as an ameliorant, is probably not effective, but in other soils poor in sodium ions it may be advisable. Previously, the use of sodium humate as a biostimulator of native soil microbiota in various types of pollution was studied [4,23]. In this study, a 1% solution of sodium humate “GUMI-30” was added into the soil for remediation (Ufa, Republic of Bashkortostan, Russia).

The microbiological preparation of Baikal EM-1 contained 60 strains of beneficial microorganisms. The composition of this drug includes bacterial strains (*Paenibacillus pabuli*, *Azotobacter vinelandii*, *Lactobacillus casei*, *Clostridium limosum*, *Cronobacter sakazakii*, *Rhodotorulla mucilaginosa*, *Cryptococcus*), hybrid yeast (*Saccharomyces*, *Candida lipolitica*, *Candida norvegensis*, *Candida guilliermondii*), and fungi (*Aspergillus*, *Penicillium*, *Actinomycetales*). Fertilizer Baikal EM-1 produced by LLC “Scientific and Production Association EM-CENTER” was used in this study (Ulan-Ude, Republic of Buryatia, Russia). A 1% water solution of this drug was introduced into the soil.

To simulate soil contamination with oil, 200 g of air-dried soil was weighed and placed in a plastic vessel in triplicate (Figure 1). Oil was introduced into the moistened soil (40% of the field humidity) and thoroughly mixed. Biochar was crushed in a mortar to a fraction size of 0.2–0.5 cm and mixed with contaminated soil. Percentage solutions were prepared from sodium humate and Baikal EM-1, which moistened the soil before applying oil. Incubation of vegetative vessels was carried out for 30 days while maintaining optimal temperature (24–25 °C) and soil moisture (35%).

At the end of the exposure period, an assessment of the ecological state of the soil was carried out. To do this, residual oil content in the soil was determined by extraction with carbon tetrachloride [24]. The biological properties of soils were evaluated according to the indicators presented in Table 2. 

To assess the ecological state of the soil in terms of soil enzymatic activity, an integral indicator of soil enzymatic activity (IIFA) was calculated using Equations (1)–(3):(1)$$E_{1\;}=\frac{E_{x}}{E_{\mathrm{cont}}}\times 100\%$$
where E_1_ is the relative score of enzyme activity; E_x_ is the actual value of enzyme activity; and E_cont_ is the control value of enzyme activity.
(2)$$E_{\mathrm{av}}=\frac{E_{1}+E_{2}+⃯+E_{n}}{N}$$
where E_av_ is the average estimated score of enzyme activity; E_1_…E_n_ is the relative score of enzyme activity; and N is the amount of enzyme activity.

The integral index of enzymatic activity of soil (IIEA) is calculated by Equation (3):(3)$$\mathrm{IIEA}=\frac{E}{E_{\mathrm{ref}}}\times 100\%$$
where E is the average estimated score of enzyme activity and E_ref_ is the control of enzyme activity.

The integral index of enzymatic activity of soil (IIEA) yielded an integral assessment of the condition of soils after any chemical pollution. For the calculation of IIEA, the value of each of the above indicators in the control (in unpolluted soil) was taken as 100%. The percentages in other experimental variants (in polluted soil) were expressed as a percentage relative to control. For the IIEA condition, the maximum value of each index (100%) was chosen from array data, and in reference to the value of this index, was expressed for other variants of experiments in Equation (3). The relative values of several indicators, namely the activity of catalase, dehydrogenases, invertase, phosphatase, and urease, were summed.

An analysis of the rate of variation (standard deviation) at *p* ≤ 0.05 was conducted to determine the reliability of the results. Data were means of triplicate. Statistical data processing was carried out using Statistica 12.0 and the Python 3.6.5 Matpolib package.

## 3. Results

Residual oil content in soils after application of biochar, sodium humate, and Baikal EM-1 is shown in Figure 2. When applying biochar, the highest decomposition efficiency was established in Haplic Cambisols and Haplic Arenosols—12 and 94%, respectively, of the initial content. In Haplic Chernozem under these incubation conditions (duration of 30 days, optimal humidity and air temperature), the oil content did not change compared to the background. The introduction of sodium humate contributed to a decrease in the concentration of petroleum hydrocarbons in Haplic Chernozem, Haplic Cambisols, and Haplic Arenosols by 25, 26, and 93% of the initial content, respectively. The microbiological preparation of Baikal EM-1 contributed to oil decomposition in Haplic Chernozem, Haplic Cambisols, and Haplic Arenosols by 54, 26, and 29% compared to the background content, respectively.

The effect of ameliorants on oil decomposition in soils is presented in the form of a series (in % relative to the control) for each studied soil type.
Haplic Chernozem: sodium humate > Baikal EM-1 > biochar
Haplic Cambisol: Baikal EM-1 = sodium humate > biochar
Haplic Arenosols: biochar > sodium humate > Baikal EM-1

*Changes in the physicochemical properties of soils after the introduction of biochar*, *sodium humate*, *and Baikal EM-1.* The concentration of easily soluble salts also plays an important role by influencing the vital activity of microorganisms (Table 3).

Taking into account the soil texture and content of organic matter, pH depended on the soil type; for light loamy soils as Haplic Arenosols, pH values > 7 were noted, and for clay loam soils (Haplic Chernozem and Haplic Cambisols), pH ≤ 7 levels were noted (Figure 1). Among the ameliorants, the introduction of biochar caused an increase in pH from 5.3 (Haplic Cambisols) to 8.2 (Haplic Arenosols). A significant decrease in pH was observed after the introduction of sodium humate (Haplic Chernozem) and Baikal EM-1 (Haplic Cambisol and Haplic Arenosols).

The content of easily soluble salts in the control soil without oil was different depending on soil type. In Haplic Chernozem and Haplic Cambisols, the content of easily soluble salts was 0.27 and 0.15 mg/kg, and in Haplic Arenosol, the content was 8.35 mg/kg. The increased salt content in Haplic Arenosols in comparison with other soils is due to the soil texture and the area of the inner surface of the soils. This value does not exceed the permissible units of easily soluble salts in the soil.

The content of easily soluble salts in oil-contaminated Haplic Chernozem did not change with the addition of sodium humate and Baikal EM-1. When applying biochar, the content of easily soluble salts increased by 96% relative to oil-contaminated soil without the introduction of ameliorants. In oil-contaminated Haplic Cambisols, the content of easily soluble salts with the addition of biochar and Baikal EM-1 increased by 83 and 58%, respectively. The salt content in the oil-contaminated Haplic Arenosols, which have a light soil texture, increased eight-fold after the addition of Baikal EM-1. At the same time, the excess content of easily soluble salts was still not established, but on the contrary, the lack of salt content in the soil was obvious. 

The change in the redox potential (RP) was not as significant as the change in the content of easily soluble salts. Reduction potential (RP) values in uncontaminated Haplic Chernozem, Haplic Cambisols, and Haplic Arenosols were 273, 288, and 223 mV, respectively. Redox potential values in clay loam soils (Haplic Chernozem and Haplic Cambisols) were 18 and 23% higher than in Haplic Arenosol. When oil was added in Haplic Chernozem and Haplic Cambisols, the values of RP did not change, and in Haplic Arenosol, values increased by 10%.

The Increase in RP is a consequence of the activation of microorganisms decomposing petroleum hydrocarbons [25]. The addition of biochar and sodium humate increased the RP of Haplic Chernozem by 19 and 34% relative to oil pollution and contributed to the intensification of oil decomposition by soil microorganisms. In Haplic Chernozem, the introduction of “Baikal EM-1” did not affect the RP. In Haplic Cambisols, on the contrary, the addition of biochar and Baikal EM-1 reduced the RP by 15 and 17%, respectively, and sodium humate did not affect the RP of the soil. In oil-contaminated Haplic Arenosols, only biochar caused a decrease in RP by 16% compared to oil pollution (8% lower than the control). This indicates low oxygen saturation of the soil. 

*Changes in the activity of oxidoreductases after the introduction of biochar, sodium humate, and Baikal EM-1.* Changes in the enzymatic activity of soils were assessed by the activity of enzymes of the oxidoreductase and hydrolase class [26]. In oil-contaminated soil, catalase activity decreased by 31 and 25% compared to the background in Haplic Chernozem and Haplic Cambisols. In Haplic Arenosols, enzyme activity was stimulated by 62%. The activity of dehydrogenases in all oil-contaminated soils did not significantly differ from the control. The activity of oxidoreductases during the introduction of biochar, sodium humate, and Baikal EM-1 for the studied soil types varied ambiguously (Figure 3). 

The activity of catalase of Haplic Chernozem when introducing ameliorants into oil-contaminated soil decreased by 15–36%. A similar trend was found for catalase activity in Haplic Cambisols with the introduction of biochar and sodium humate. Baikal EM-1 had no effect on the enzymatic activity under study in Haplic Cambisols. The introduction of biochar, sodium humate, and Baikal EM-1 stimulated catalase activity by 267, 64, and 63% of the control, respectively. The activity of dehydrogenases in Haplic Chernozem increased only with the introduction of biochar by 25% of the control. The remaining ameliorants had no effect on the activity of Haplic Chernozem dehydrogenases compared with the control. The dehydrogenases activity of Haplic Cambisols and Haplic Arenosols during the introduction of ameliorants into oil-contaminated soil was stimulated by 53–245% compared to the control.

The effect of ameliorants on the activity of oxidoreductases is presented in the form of a series (in % relative to the control) for each studied soil type.
Haplic Chernozem: biochar > sodium humate > Baikal EM-1
Haplic Cambisols: biochar > sodium humate = “Baikal EM-1”
Haplic Arenosols: “Baikal EM-1” > sodium humate > biochar

*Changes in the activity of hydrolases after the introduction of biochar, sodium humate, and Baikal EM-1.* The activity of soil hydrolases (urease, invertase, and phosphatase) varied differently (Figure 4). In oil-contaminated soil, the activity of invertase and phosphatase did not significantly differ from the control. The urease activity of oil-contaminated Haplic Chernozem and Haplic Cambisols was stimulated by 50 and 49%, respectively, and Haplic Arenosol was inhibited by 23%.

In Haplic Chernozem, the activity of invertase and urease increased relative to the control when adding sodium humate by 14–61%. Phosphatase activity did not change with the introduction of ameliorants.

In Haplic Cambisols, invertase activity was stimulated by the addition of sodium humate by 19%, and urease activity was stimulated by the addition of biochar and Baikal EM-1 by 250 and 105% of the control, respectively. Phosphatase activity during the introduction of Baikal EM-1 was inhibited by 18%.

The invertase activity of Haplic Arenosols was stimulated by the introduction of biochar and Baikal EM-1 by 13 and 70% of the control, respectively. Urease activity was stimulated by 49% when introducing biochar into Haplic Arenosols. Phosphatase activity did not change with the introduction of ameliorants.

The effect of ameliorants on the activity of hydrolases is presented in the form of a series (in % relative to the control) for each studied soil type.
Haplic Chernozem: sodium humate > Baikal EM-1 > biochar
Haplic Cambisols: biochar > Baikal EM-1 > sodium humate
Haplic Arenosols: biochar > “Baikal EM-1” > sodium humate

*Integral indicator of the enzymatic activity of soils*. Figure 5 shows an integral indicator of the enzymatic activity of soils (IIEA), calculated according to Equation (3).

Biochar in oil-contaminated Haplic Cambisols and Haplic Arenosols stimulated IIEA by 60 and 43% of the control, respectively. Sodium humate stimulated IIEA in Haplic Chernozem, Haplic Cambisols, and Haplic Arenosols by 10, 31, and 28% of the control, respectively. The microbiological preparation of “Baikal EM-1” stimulated IIEA in Haplic Cambisols and Haplic Arenosols by 37 and 53%, respectively, relative to the control.

## 4. Discussion

The enzymatic activity (catalase, dehydrogenases, invertase, urease, and phosphatase) are the most informative and sensitive indicators when soils are contaminated with petroleum products [2,5,27,28]. To assess the informativeness of enzymes, the correlation coefficients between the residual oil content and the change in biological activity were calculated. The most informative indicator when using ameliorants in Haplic Chernozem and Haplic Cambisols is the activity of dehydrogenases, with correlation coefficients R = −0.97 and R = −0.70, respectively. For Haplic Arenosols, phosphatase activity is represented by R = −0.55. Such differences in the information content of enzymes for different types of soils are due to differences in the physicochemical properties of these soils (Table 3). Correlation coefficients were calculated for each ameliorant with oil content in the soil (Table 4).

According to the calculated correlation coefficients between oil content and the activity of soil enzymes, it was found that when applying biochar, the most informative enzyme was invertase activity (R = −0.89); for sodium humate, urease activity (R = −0.86); for Baikal EM-1, invertase activity (R = −0.99) and phosphatase activity (R = −0.95). Without the introduction of ameliorants, the closest relationship was found with catalase activity (R = −0.96). Thus, the most informative indicator for the remediation of Haplic Chernozem and Haplic Cambisols is the activity of dehydrogenases, and for Haplic Arenosols, this indicator is the activity of phosphatase.

Oil significantly violates the water–salt regime of the soil. Among the most informative indicators of enzymatic activity, urease activity is distinguished for Haplic Chernozem and Haplic Arenosols with oil pollution. Urease activity characterizes the course of the nitrogen cycle in the soil. In oil-contaminated soil, when stimulating enzymatic activity, a decrease in oil content is observed, which leads to an alignment of the C:N ratio to a favorable 10:1. Previously, the informativeness and sensitivity of urease were established in case of oil contamination of sandy loam soil at the University of Krakow (Poland) after the application of microbial preparation of ZB-01 with nitrogen fertilizer [29]. The effectiveness of restoring the biological properties of the soil after the introduction of ameliorants is presented in the series: Haplic Cambisols > Haplic Arenosol > Haplic Chernozem.

Such differences in the rate of oil decomposition during the introduction of ameliorants are mainly due to the physicochemical properties of soils, namely the content of organic matter, pH, and soil structure [30]. Differences in the physicochemical and biological properties of the studied soils determine their different resistance to pollutants of a particular nature, such as heavy metals and oil derivatives [2,5,27]. The low recovery rate of Haplic Chernozem is probably due to its physicochemical properties, such as soil texture, organic matter content, and the reaction of the soil environment (pH). The reaction of the soil environment (pH) influences the process of oil decomposition by the native microbiota [31]. Easily soluble salts contain cations and anions of elements necessary for the vital activity of soil biota and plants [32]. At the same time, the redox potential of soils is of considerable importance. It was previously established that these physical properties determine the speed and degree of soil recovery under various anthropogenic influences [33]. This is due to aeration conditions and the vital activity of the soil microbiota [34]. With good soil aeration, RP values vary in the range of 350 to 750 mV. At the same time, according to Figure 3, oil decomposition during the introduction of biochar was the most intense in Haplic Arenosols. Despite the low oxygen supplementation, this is sufficient to stimulate the activity of the soil microbiota and oil decomposition in Haplic Arenosols when applying biochar. Thus, it is soil dispersion that influences the effectiveness of a particular ameliorant in the soil. As a result of a significant increase in soil reaction (pH), the decomposition of petroleum hydrocarbons may slow down. The introduction of Baikal EM-1 helps to reduce the pH < 7.6–7.7 and increase the RP of Haplic Arenosols, which reduces the rate of oil decomposition. It is obvious that the process of restoring the soil structure disturbed by oil pollution depends on the soil texture composition [35]. Light loamy soils such as Haplic Arenosols contain much less clay and silt in their composition than clay loam soils. The different dispersion and the area of the inner surface of the soils cause a greater efficiency in introducing biochar into Haplic Arenosols than biostimulators, such as sodium humate and Baikal EM-1. Such differences in the information content and sensitivity of the indicators are due to the physicochemical properties of the soil and soil texture. In Haplic Arenosols and Haplic Cambisols, which have a lower phosphorus content and a lighter soil texture in comparison with Haplic Chernozem, phosphatase activity is the most informative. In Haplic Arenosols and Haplic Cambisols, the most sensitive indicator is the length of the radish roots. Therefore, the recovery period after oil pollution in Haplic Arenosols occurs faster than in Haplic Chernozem [36,37]. In Haplic Chernozem, the phosphorus content depends on the cultivation technology and the fertilizers used [38,39,40]. An increase in the fraction of mobile phosphorus is typical during plowing, as a result of which its content decreases over time [41,42].

## 5. Conclusions

As a result of studying the remediation of oil pollution of ordinary and Haplic Arenosol, Haplic Cambisols with sodium humate, biochar, and Baikal EM-1, it was found that the effectiveness of these substances on the biological properties of soils and the residual oil content differed. The most effective ameliorants for restoring the ecological state of Haplic Cambisols after oil pollution was biochar, for Haplic Arenosols, this was sodium humate, and for Haplic Chernozem, the effectiveness of biochar and sodium humate did not differ. The most informative indicator for the remediation of Haplic Chernozem and Haplic Cambisols is the activity of dehydrogenases, and for Haplic Arenosols, this is the activity of phosphatase. The results of the study should be used in biomonitoring the ecological state of oil-contaminated soils after bioremediation.

## Figures and Tables

**Figure 1 toxics-11-00355-f001:**
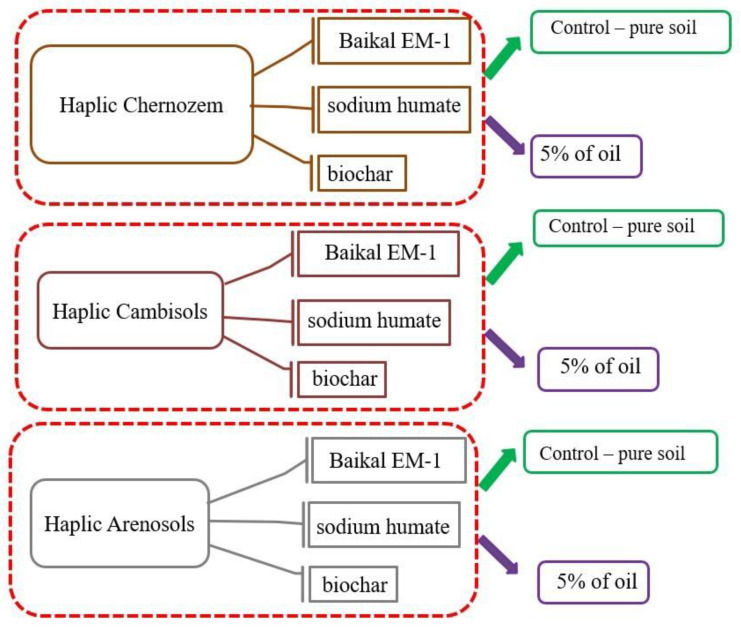
Scheme of the experiment on bioremediation of oil-contaminated soils in the South of Russia.

**Figure 2 toxics-11-00355-f002:**
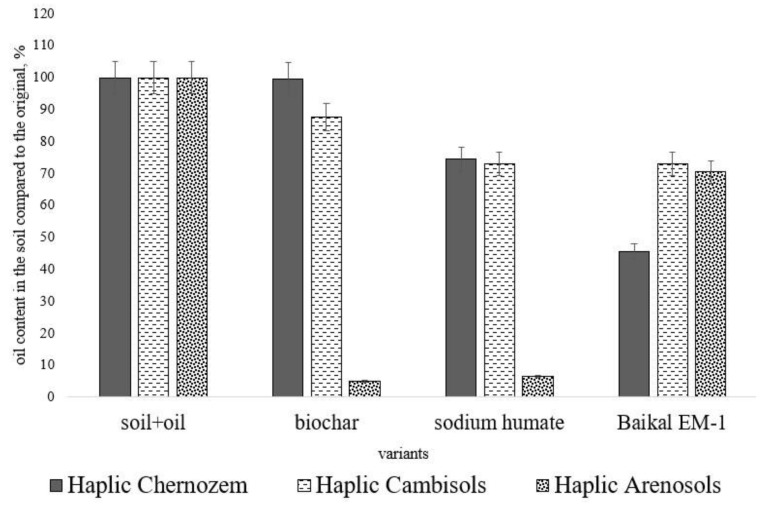
Residual oil content in the soils of the South of Russia after bioremediation, % of the initial content in the soil without ameliorants.

**Figure 3 toxics-11-00355-f003:**
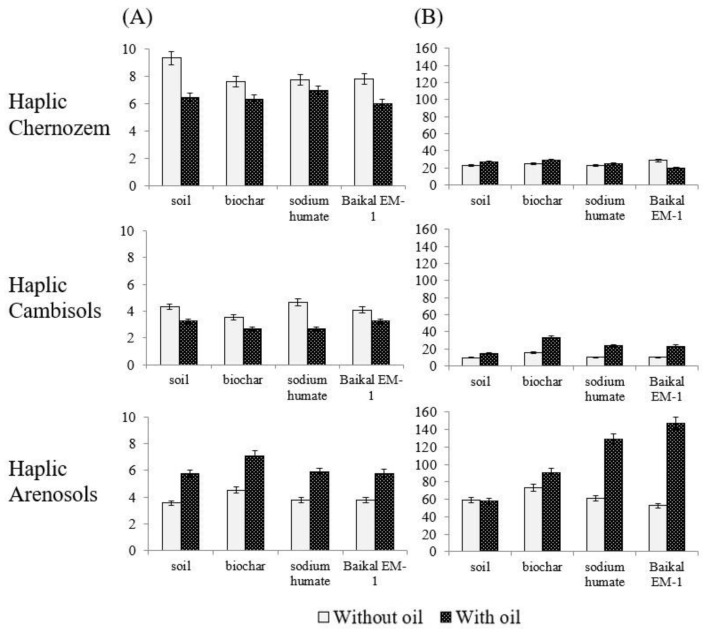
Changes in the activity of oxidoreductases in soils after the introduction of biochar, sodium humate, and Baikal EM-1: (**A**) activity of catalase (mL O_2_/1 g of soil/1 min); (**B**) activity of dehydrogenases (mg TPF/10 g of soil/24 h).

**Figure 4 toxics-11-00355-f004:**
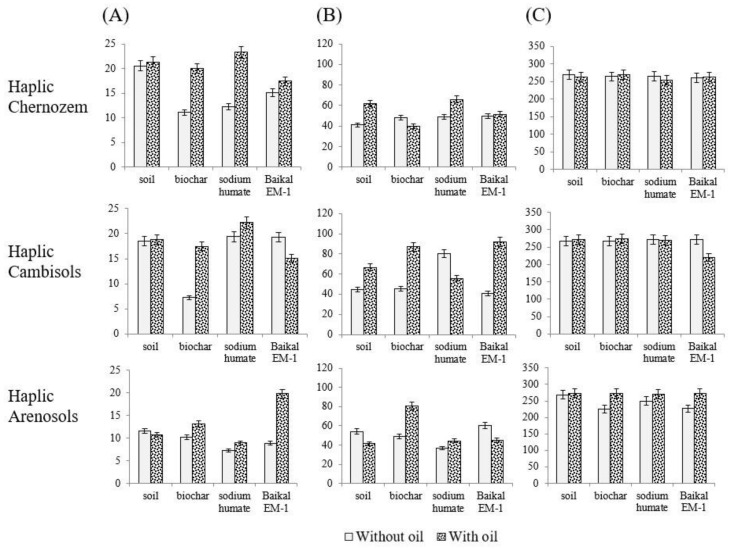
Changes in the activity of hydrolases in soils after the introduction of biochar, sodium humate, and Baikal EM-1: (**A**) invertase (mg glucose/10 g of soil/24 h); (**B**) urease (mg NH_3_/10 g of soil/24 h); (**C**) phosphatase (1 µg p-nitrophenol/1 g of soil/h).

**Figure 5 toxics-11-00355-f005:**
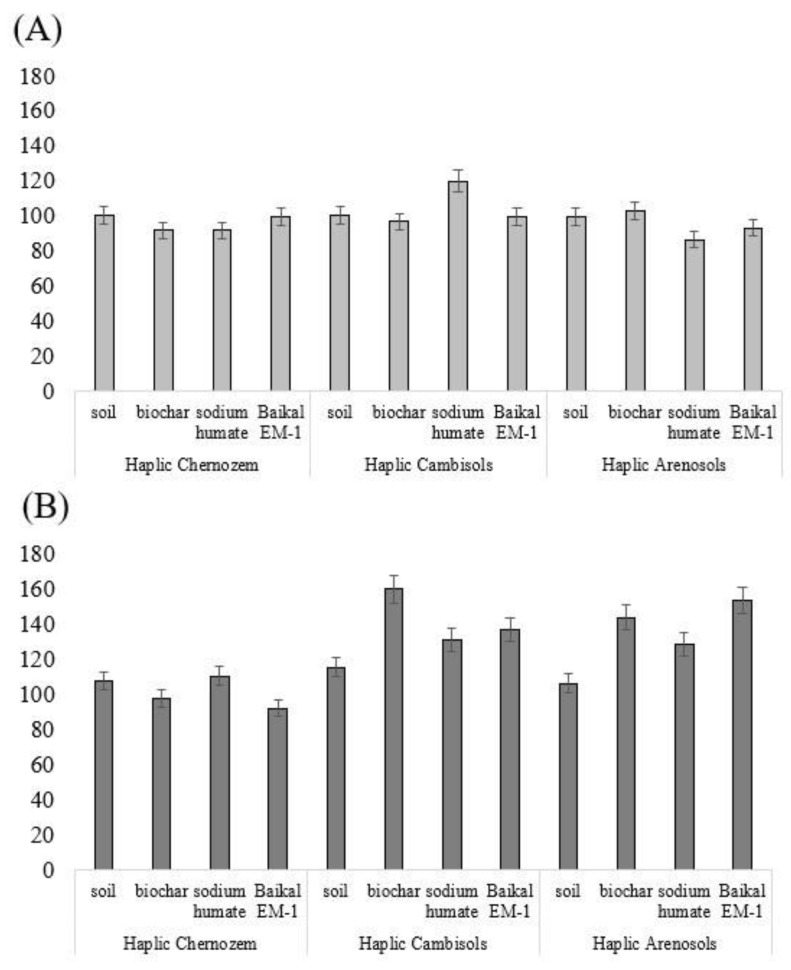
The integral indicator of the enzymatic activity of soils in the South of Russia during remediation with biochar, sodium humate, and Baikal EM-1: (**A**) without oil; (**B**) with oil.

**Table 1 toxics-11-00355-t001:** Soil sampling points and physicochemical properties.

Soil Type	Sampling Location	Coordinates	Land Use Type	Soil Texture	Soil Organic Carbon, %	pH
Haplic Chernozem	Botanical Garden of SFU (Rostov-on-Don)	47°14′17.54 N 39°38′33.22 E	arable land	heavy loamy	3.7 ± 0.2	7.1 ± 0.5
Haplic Arenosols	Ust’-Donetsk region, (Rostov region)	47°46.015 N 40°51.700 E	arable land	light loamy	1.8 ± 0.1	5.1 ± 0.3
Haplic Cambisols	Nickel village, (Rep. Adygea)	44°10.649 N 40°9.469 E	hornbeam-beech forest	heavy loamy	2.3 ± 0.1	8.2 ± 0.3

**Table 2 toxics-11-00355-t002:** Physicochemical and biological indicators of the ecological state of the soil.

№	Indicator	Method of Measurement	Units of Measurement
Physical, physico-chemica,l and chemical indicators			
1.	Residual oil content	Extraction from the soil with carbon tetrachloride followed by determination of the quantitative content by IR spectroscopy	mg/kg
2.	pH	Potentiometric method	-
3.	The content of easily soluble salts (Cl^−^, SO^−2^_4_, HCO_3_^−^ Ca^2+^, Mg^2+^)	Conductometric method	mg/kg
4.	Redox potential	Voltammetric method	mV
Enzyme activity			
5.	Catalase activity	Gasometric method for the decomposition of hydrogen peroxide	mL O_2_/1 g /1 min
6.	Dehydrogenases activity	Reduction in tetrazolium chloride salts to triphenylformazans (TPF)	mg TPF/10 g of soil/24 h
7.	Invertase activity	Colorimetric method to change the content of reducing sugars	mg glucose/1 g of soil/24 h
8.	Urease activity	Colorimetric method according to the amount of released ammonia	мг NH_3_/10 g of soil/24 h
9.	Phosphatase activity	Colorimetric method according to the amount of released ammonia	µg p-nitrophenol/1 g of soil/1 h

**Table 3 toxics-11-00355-t003:** Changes in the physical and chemical properties of the soil (in the numerator: soil without oil; in the denominator: soil with oil).

№	pH	SS	RP
Haplic Chernozem			
Without ameliorants	7.1 ± 0.05 7.9 ± 0.35	0.27 ± 0.01 0.16 ± 0.01	273 ± 3.00 271 ± 9.00
+biochar	7.4 ± 0.20 7.2 ± 0.04	0.20 ± 0.01 0.32 ± 0.01	288 ± 2.50 325 ± 8.00
+sodium humate	6.4 ± 0.36 6.8 ± 0.05	0.11 ± 0.01 0.16 ± 0.04	371 ± 1.00 362 ± 2.00
+Baikal EM-1	7.0 ± 0.22 7.2 ± 0.15	0.15 ± 0.02 0.11 ± 0.03	344 ± 4.50 269 ± 2.50
Haplic Cambisols			
Without ameliorants	5.1 ± 0.08 5.3 ± 0.07	0.15 ± 0.02 0.18 ± 0.04	288 ± 3.90 289 ± 2.50
+biochar	5.9 ± 0.01 5.5 ± 0.30	0.13 ± 0.03 0.28 ± 0.08	343 ± 2.00 246 ± 4.50
+sodium humate	5.2 ± 0.05 5.8 ± 0.09	0.14 ± 0.01 0.12 ± 0.01	344 ± 6.50 273 ± 5.00
+Baikal EM-1	5.7 ± 0.37 5.6 ± 0.15	0.17 ± 0.02 0.24 ± 0.01	355 ± 7.00 238 ± 2.50
Haplic Arenosols			
Without ameliorants	8.2 ± 0.13 8.0 ± 0.26	8.35 ± 0.25 0.05 ± 0.01	223 ± 7.50 247 ± 6.50
+biochar	8.3 ± 0.02 8.1 ± 0.02	0.15 ± 0.01 0.15 ± 0.08	217 ± 2.00 205 ± 2.00
+sodium humate	8.2 ± 0.10 7.6 ± 0.60	5.69 ± 0.31 0.05 ± 0.01	242 ± 2.00 238 ± 6.00
+Baikal EM-1	7.7 ± 0.60 7.6 ± 0.69	6.34 ± 0.01 4.00 ± 0.10	268 ± 2.50 267 ± 3.00

Note: SS—content of easily soluble salts, mg/kg; RP—redox potential, mV.

**Table 4 toxics-11-00355-t004:** Correlation coefficients between oil content and soil enzyme activity values.

Ameliorant	A_cat_	A_deh_	A_inv_	A_ur_	A_phos_
biochar	−0.16	0.71 *	−0.89 **	0.95 *	0.98 *
sodium humate	−0.68 *	0.52	−0.58 *	−0.87 **	1.00 *
Baikal EM-1	−0.87 **	−0.81 **	−1.00 **	0.95 *	−0.96 **

Note: A_cat_—activity of catalase; A_deh_—activity of dehydrogenases; A_inv_—activity of invertase; A_ur_—activity of urease; A_phos_—activity of phosphatase. *—*p* < 0.05, **—*p* < 0.01

## Data Availability

Not applicable.

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
