# Peer review of "Enzymatic Assessment of the State of Oil-Contaminated Soils in the South of Russia after Bioremediation"

_toxics, 2023, doi:10.3390/toxics11040355_

Round 1

Reviewer 1 Report

The article in general needs a revision of the English form, some sentences are not fluent and not very understandable.  There are some comments in the attached file to improve the level of the article, especially in the discussion part where you have to add extra phrases about biochar, since it turned out to be the best product. Materials and methods can be rewritten better to make them more understandable. The aims of the work should be moved as suggested in the attached file. In the discussion there are some parts to move in the results as suggested in the attached file. In general I recommend making the whole text more linear and understandable to make reading better and easier understanding. The graphs are too small and the error bars are not visible.

That said, I think that after performing the recommended revisions the article is publishable.

Author Response

Our team of authors thanks you for your comments. Replies to the comments helped to significantly improve the quality of the work.

Thank you!

Reviewer 2 Report

In the manuscript the results of 30 days long incubation experiment with three different soils  contaminated with oil, which are remediated with three products are presented. However, the subject undertaken is interesting and important the manuscript has many drawbacks.

General comments:

First of all English needs correction and improvement, then the names of the soil used in the experiment should be kept the same throughout the whole manuscript, from time to time authors use the direct translation of the soil names from their language which is confusing. Furthermore it should be emphasized that the results are from the incubation experiment thus they are only preliminary to be proved in further study. In the title it should be mentioned that there are results of the incubation experiment and not in the field in South of Russia as the title suggests. Generally the article is not well organized and there are sentences with the literature citations or expressing opinions put in Results or Methods section instead of Discussion. Another option is to reorganize the article by combining Results and Discussion sections, but not something in between as it is now.

Detailed comments are listed below:

Line 68: There are soils belonging to three main groups according to WRB soil classification. Granulometric composition ought to be called- texture and organic carbon content is given without a unit. As a texture in Table 1 it is listed heavy loam or light loam and in the text it is clay loam (chernozem).

Lines 69- 81: Use, please names according to WRB throughout the whole article. How many treatments there were in total? What it means A 0-10? Is this a depth 0-10 cm of A horizon? Were all samples taken from 0-10 cm depth? There is no such information about other than the first object. I do not understand why chernozem is called Arenosol  (the second treatment) they are completely different soils.

Lines 82-84: Information given in these lines have nothing to do with materials and methods and should be moved to Discussion section.

Line 85: Eliminate this sentence, please.

Line 88: What do you mean by mass fraction? Explain please

Line 91: How much soil was in each vessel?

Line 98: What does  ‘OOO’ mean?

Line 115: What are the contents of heavy metals in the  sodium humate? Are they below permissible values? It does not make sense to remediate soil from one pollutant by introducing another.

Line 129: pH is soil reaction, not acidity

Line 131-132: Move these sentences with the literature citation to the Discussion section, please

Table 3: What easily soluble salts were measured?

Line 139: There is not Eref in the equation 1. There is Econt which is not explained.

Line 144-145: What is the source of the reference value? What do you mean by array data in this context?

Line 159: A verb is missing in the sentence, correct it please.

In Figure 2, the first treatment is control, I suppose

Lines 175-182: Move these sentences to the Discussion section as they contain the possible explanation of the results, however it is not enough to say that the different soil reaction to remediation  were due to their different properties but the explanation in what way listed properties affect the decomposition of oil.

Line 185:  There is RP in the headings of the column, not ORP as it is explained under the Table 3

Line 188: No pH values in Figure 1

Line 194: Use exclusively WRB soil names, please

Line 207: Move this to Discussion and develop, by giving examples of such cations, please

Lines 213-214: Move explanation to Discussion, please.

Lines 225-233: as above

Figures 3, 4 and 5 are of poor quality.

Lines 287-290: The correlation coefficients are calculated for the correlation between enzyme activity and what? Oil content? It is not clear. Moreover, the correlation coefficients are extremely high, even equal to 1, or -1, describing correlation between two different properties? It is impossible!

Line 310: A ratio between C and N equal to 10 is the most favourable.

Lines 316: This sentence as the explanation is repeated a few times in the text, always without further explanation. Elaborate it, please, giving some examples.

Line 329: Explain the effect of the length of radish roots on the phosphatase activity, please

Author Response

Author responses to reviewer (2) comments

(agriculture- 2260949)

To Reviewer #2

In the manuscript the results of 30 days long incubation experiment with three different soils  contaminated with oil, which are remediated with three products are presented. However, the subject undertaken is interesting and important the manuscript has many drawbacks.

General comments:

First of all English needs correction and improvement, then the names of the soil used in the experiment should be kept the same throughout the whole manuscript, from time to time authors use the direct translation of the soil names from their language which is confusing. Furthermore it should be emphasized that the results are from the incubation experiment thus they are only preliminary to be proved in further study. In the title it should be mentioned that there are results of the incubation experiment and not in the field in South of Russia as the title suggests. Generally the article is not well organized and there are sentences with the literature citations or expressing opinions put in Results or Methods section instead of Discussion. Another option is to reorganize the article by combining Results and Discussion sections, but not something in between as it is now.

Detailed comments are listed below:

Line 68: There are soils belonging to three main groups according to WRB soil classification. Granulometric composition ought to be called- texture and organic carbon content is given without a unit. As a texture in Table 1 it is listed heavy loam or light loam and in the text it is clay loam (chernozem).

Response: Text have been corrected.

Lines 69- 81: Use, please names according to WRB throughout the whole article. How many treatments there were in total? What it means A 0-10? Is this a depth 0-10 cm of A horizon? Were all samples taken from 0-10 cm depth? There is no such information about other than the first object. I do not understand why chernozem is called Arenosol  (the second treatment) they are completely different soils.

Response: Text have been corrected.

Lines 82-84: Information given in these lines have nothing to do with materials and methods and should be moved to Discussion section.

Response: Text have been removed in Discussion.

Line 85: Eliminate this sentence, please.

Response: Sentence was deleted.

Line 88: What do you mean by mass fraction? Explain please

Response: Text have been corrected.

Line 91: How much soil was in each vessel?

Response: This information is on line 118.

Line 98: What does  ‘OOO’ mean?

Response: Text have been corrected.

Line 115: What are the contents of heavy metals in the sodium humate? Are they below permissible values? It does not make sense to remediate soil from one pollutant by introducing another.

Response: Sodium humate contains insignificant concentrations of lead and cadmium, the amount of which in 1% solution is reduced by 1000 times. This dose does not have a toxic effect on the soil at this dilution.

Line 129: pH is soil reaction, not acidity

Response: Text have been corrected.

Line 131-132: Move these sentences with the literature citation to the Discussion section, please

Response: Text have been moved.

Table 3: What easily soluble salts were measured?

Response: The method for determining salts is given in Table 2.

Line 139: There is not Eref in the equation 1. There is Econt which is not explained.

Response: Text have been corrected.

Line 144-145: What is the source of the reference value? What do you mean by array data in this context?

Response: This is control value/ Text have been corrected.

Line 159: A verb is missing in the sentence, correct it please.

Response: Sentences have been corrected.

In Figure 2, the first treatment is control, I suppose

Response: No, this is oil polluted soil (i.e polluted soil without ameliorants).

Lines 175-182: Move these sentences to the Discussion section as they contain the possible explanation of the results, however it is not enough to say that the different soil reaction to remediation  were due to their different properties but the explanation in what way listed properties affect the decomposition of oil.

Response: Text have been moved to Discussion.

 Line 185:  There is RP in the headings of the column, not ORP as it is explained under the Table 3

Response: Text have been corrected.

Line 188: No pH values in Figure 1

Response: Text have been corrected

Line 194: Use exclusively WRB soil names, please

Response: Text have been corrected

Line 207: Move this to Discussion and develop, by giving examples of such cations, please

Response: Text have been moved to Discussion.

Lines 213-214: Move explanation to Discussion, please.

Response: Text have been corrected

Lines 225-233: as above

Response: Text have been corrected

Figures 3, 4 and 5 are of poor quality.

Response: Figures have been corrected.

Lines 287-290: The correlation coefficients are calculated for the correlation between enzyme activity and what? Oil content? It is not clear. Moreover, the correlation coefficients are extremely high, even equal to 1, or -1, describing correlation between two different properties? It is impossible!

Response: Yes, calculation of correlation coefficients between oil content and enzyme activity (this is written in the title of Table 2). Added to article text. Text have been corrected

Line 310: A ratio between C and N equal to 10 is the most favourable.

Response: Corrected

Lines 316: This sentence as the explanation is repeated a few times in the text, always without further explanation. Elaborate it, please, giving some examples.

Response: Text have been corrected.

Line 329: Explain the effect of the length of radish roots on the phosphatase activity, please

Response: This part of the text does not say that these indicators are related, it is written here which indicator in Haplic Arenosols and Haplic Cambisols is the most informative (phosphatase), and which is the most sensitive (root length). This sentence will be rewritten.

Round 2

Reviewer 1 Report

dear authors,

the manuscript has improved considerably compared to the first version. 

As far as I am concerned, it is now publishable. 

Excellent work, and good luck for your future research

Author Response

Author responses to reviewer (1) comments

(agriculture- 2260949)

To Reviewer #1

dear authors,

the manuscript has improved considerably compared to the first version. 

As far as I am concerned, it is now publishable. 

Excellent work, and good luck for your future researchе

Response: Our team of authors thanks you for such a detailed review, which helped to significantly improve the text of the article.

Reviewer 2 Report

The manuscript has been corrected and much improved. I remains to change granulometric composition to soil texture throuout the whole manuscript and explain what easily soluble salts are that contents you examined. Besides that evrything is ok.

Author Response

Author responses to reviewer (2) comments

(agriculture- 2260949)

To Reviewer #2

The manuscript has been corrected and much improved.

I remains to change granulometric composition to soil texture throuout the whole manuscript and explain what easily soluble salts are that contents you examined.

Besides that everything is ok.

Response: Text have been corrected: to change “granulometric composition” to “soil texture” throughout the whole manuscript. We determined the content of the following easily soluble salts: Cl-, SO-24 , HCO3- Ca2+, Mg2+.

Thank you very much.
